# Domain-dependent strain and stacking in two-dimensional van der Waals ferroelectrics

Chuqiao Shi[1], Nannan Mao[2,3], Kena Zhang [4], Tianyi Zhang [2], Ming-Hui Chiu[2], Kenna Ashen[5], Bo Wang [6], Xiuyu Tang[5], Galio Guo[1], Shiming Lei[7], Longqing Chen [6], Ye Cao [4], Xiaofeng Qian [5,8,9], Jing Kong [2] & Yimo Han [1] ✉

Van der Waals (vdW) ferroelectrics have attracted significant attention for their potential in next-generation nano-electronics. Two-dimensional (2D) group-IV monochalcogenides have emerged as a promising candidate due to their strong room temperature in-plane polarization down to a monolayer limit. However, their polarization is strongly coupled with the lattice strain and stacking orders, which impact their electronic properties. Here, we utilize four-dimensional scanning transmission electron microscopy (4D-STEM) to simultaneously probe the in-plane strain and out-of-plane stacking in vdW SnSe. Specifically, we observe large lattice strain up to 4% with a gradient across ~50 nm to compensate lattice mismatch at domain walls, mitigating defects initiation. Additionally, we discover the unusual ferroelectric-to-antiferroelectric domain walls stabilized by vdW force and may lead to anisotropic nonlinear optical responses. Our findings provide a comprehensive understanding of in-plane and out-of-plane structures affecting domain properties in vdW SnSe, laying the foundation for domain wall engineering in vdW ferroelectrics.

Recent discoveries of two-dimensional (2D) van der Waals (vdW) ferroelectrics (FEs) are of particular interest due to the new phenomena and functionalities arising from the special atomic arrangements in combination with the ferroelectric order[1,2]. Compared with conventional FEs[3,4], vdW FE materials possess weak interlayer interaction and demonstrate atomically thin ferroelectricity, moderate band gap, and dangling-bond-free interfaces, providing opportunities to design next-generation ultra-thin memory devices and sensors[1,2,5,6]. So far, various vdW FE materials have been experimentally demonstrated, including CuInP$_2$S$_6$[7–9], Peierls-distorted WTe$_2$[10–12], In$_2$Se$_3$[13,14], bilayer h-BN[15], and group-IV monochalcogenides (abbreviated as MX, M = Ge, Sn; X = S,

Se, Te)[16–18]. Among the reported FEs, 2D MX compounds show room-temperature in-plane polarization[17,19] and giant nonlinear optical responses down to a monolayer limit[20,21], which can be manipulated by tuning various parameters including electric field and strain[22–24], making it a promising candidate for integrated device applications such as nonlinear optical and photocurrent switches[25].

Despite these promising achievements, there is a lack of understanding on the lattice distortion and heterogeneities across the domain walls at the nanometer scale, which affects critical parameters related to the phase transition, domain distribution, and domain switching of FE materials, as well as the as-resulted performance of

[1]Department of Materials Science and NanoEngineering, Rice University, Houston, TX 77005, USA. [2]Department of Electrical Engineering and Computer Science, Massachusetts Institute of Technology, Cambridge, MA, USA. [3]Department of Chemical Engineering, Massachusetts Institute of Technology, Cambridge, MA, USA. [4]Departments of Materials Science and Engineering, University of Texas at Arlington, Arlington, TX, USA. [5]Departments of Materials Science and Engineering, Texas A&M University, College Station, TX, USA. [6]Materials Research Institute and Department of Materials Science and Engineering, The Pennsylvania State University, University Park, PA, USA. [7]Department of Physics, Rice University, Houston, TX 77005, USA. [8]Department of Electrical and Computer Engineering, Texas A&M University, College Station, TX, USA. [9]Department of Physics and Astronomy, Texas A&M University, College Station, TX, USA. ✉e-mail: yimo.han@rice.edu

FE-material-based memories, sensors, and switches. MX compounds have a lattice structure analogous to phosphorene[26], where the in-plane opposite displacements between the metal and chalcogenide atoms, such as Sn and Se (Fig. 1a), are responsible for the electric polarization. Such inversion symmetry breaking associated atomic displacements are strongly coupled with lattice distortion, resulting in a strong ferroelastic effect[22,23], which may lead to complex domain structures along with built-in strain. Moreover, compared to non-vdW FEs, MX compounds hold an out-of-plane stacking order, which adds another degree of freedom and affects domain formation and switching. Note that there are two energetically favorable stacking structures of MX compounds according to theoretical predictions[27]. The bulk MX compounds (space group Pnma[28]) are commonly observed with antiferroelectric (AFE) stacking, i.e. in-plane polarization antiparallelly aligned across the neighboring vdW layers, leading to a zero net polarization. For simplicity, such a stacking is also referred to as "AB" stacking[27]. In comparison, the FE stacking (or "AC"

stacking)[27] in thin flakes contains parallel-aligned polarization that add up constructively. To date, both FE and AFE stacking orders have been reported in thin-flake samples of SnS[29] and GeSe[30] using cross-sectional TEM. Nevertheless, the domain dependent stacking order and their effects on domain wall properties are largely unexplored. The limited experimental results have motivated new approaches to revealing the strain and stacking across various domains in ultrathin MX compounds, particularly the heterogeneity manifested at distinct domain wall types.

In this work, we develop a large-scale mapping approach based on nanobeam four-dimensional scanning transmission electron microscopy[31] (4D-STEM) (Fig. 1b-e) to reveal the in-plane lattice distortion (Fig. 1f) and out-of-plane stacking information (Fig. 1g) simultaneously in vdW FE SnSe. Despite similar intertwined twin domain morphology as traditional FEs[32] (Fig. 1h-i), our results show that 2D vdW SnSe possesses unique lattice deformations at 180° ("T"-shaped) domain walls (black boxes in Fig. 1h and i) and unique FE-AFE stacking

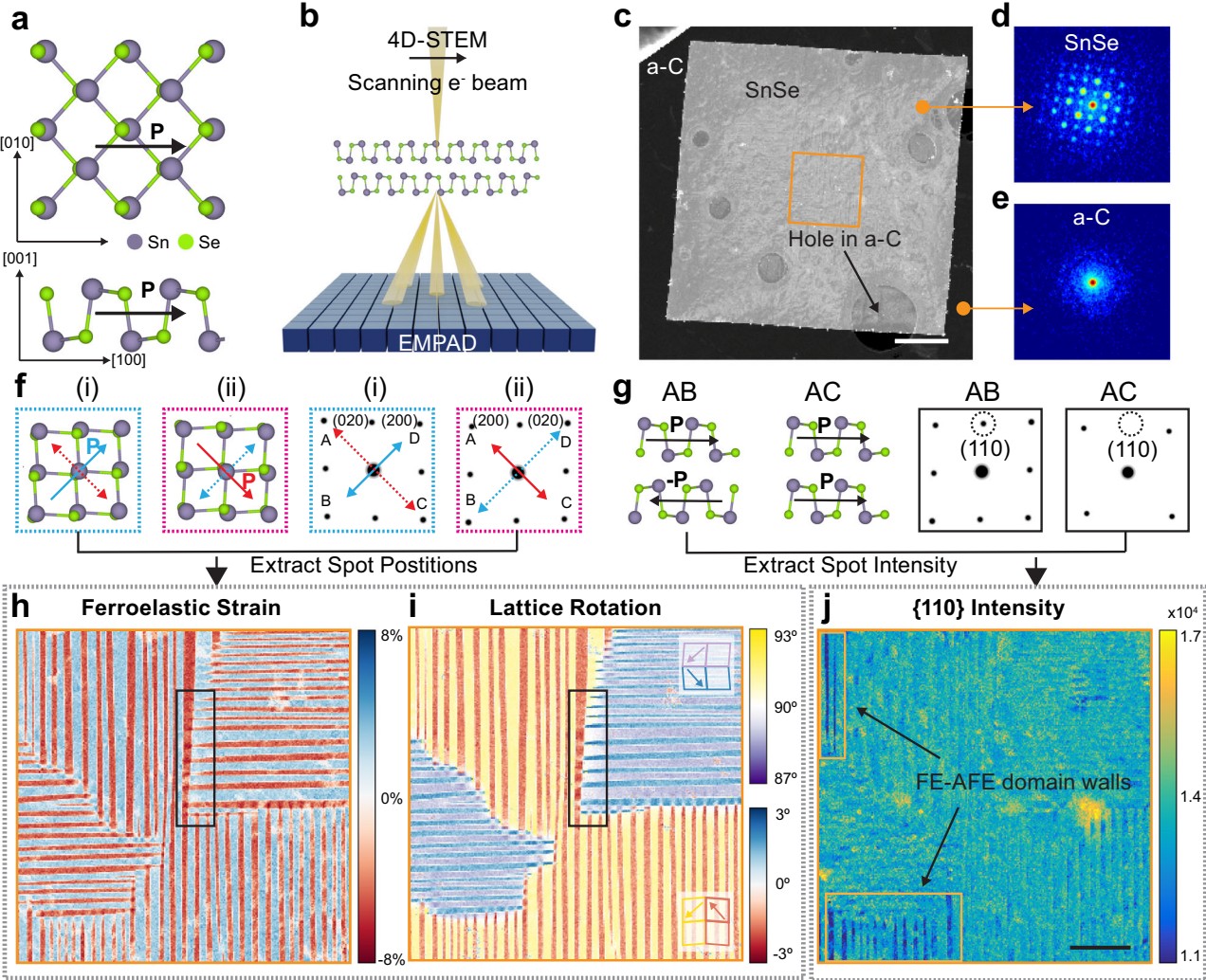

**Fig. 1 | Micrometer-scale mapping of strain and stacking by 4D-STEM. a** Lattice model of ferroelectric SnSe, where the in-plane spontaneous polarization is from the displacement of Sn and Se atoms along the [100] directions. **b** Schematic of 4D-STEM operation, where a diffraction pattern is recorded at each scan position by EMPAD. **c** ADF-STEM image reconstructed from the 4D data of a SnSe flake on a lacey carbon TEM grid. (Scale bar: 2 μm) The yellow box designates the area with intertwined domains. **d, e** Diffraction patterns from SnSe (**d**) and the amorphous carbon support (**e**). **f** Two possible spontaneous polarization directions in the SnSe flake (indicated on the lattice schematics) and their corresponding simulated diffraction patterns (boxed with the same color as the respective lattice schematic).

Solid arrows represent the armchair directions, while dotted arrows indicate zigzag directions. **g** Lattice schematics (side view) and simulated diffraction patterns of two possible stacking orders: AFE (or "AB") stacking and FE (or "AC") stacking. **h** Ferroelastic strain map of SnSe flake from the orange box in (**c**). The black box marks a superdomain boundary where stripe domains intersect perpendicularly. **i** Lattice rotation map from the same area. Insets: polarization directions in different stripe domains with the respective colors. **j** {110} intensity map indicating the out-of-plane stacking order in the sample. The yellow boxes show regions that form FE-AFE domain walls. (Scale bar: 500 nm).

transition at 90° twin walls (yellow boxes in Fig. 1j). This phenomenon remains consistent across the SnSe thin flakes we studied (Fig. S1 and Fig. S2). The findings lay the foundations for future investigation of domain wall switching and engineering in vdW FEs.

## Results

### Micrometer scale mapping of in-plane and vertical structures

Few-layer FE SnSe were grown via physical vapor deposition (PVD)[33] on mica substrate followed by a direct transfer to a TEM grid (details in Methods). We utilized 4D-STEM with the electron microscope pixel array detector (EMPAD)[34] (Fig. 1b) to collect 4D datasets on SnSe thin flakes. We observed the thin flakes with FE domains have thicknesses around ~12 nm[33] (Fig. S3). 4D-STEM scans across a large area (~10 μm) in real space (Fig. 1c) and at every scan position, the EMPAD takes a full electron diffraction pattern (Fig. 1d, e). Since we are using nanobeam mode, the diffraction patterns on 2D crystal contain well-separated diffraction spots, with the lattice information encoded in the position and intensity of these diffraction spots. Because the EMPAD has a large dynamic range of 1,000,000:1 and a single-electron sensitivity, the center of mass (CoM) measurement of the diffraction patterns provides a sub-picometer precision for the reciprocal lattice calculation[35]. In addition, the integration of the diffraction intensity of each spot from a circular masked region also offers high accuracy due to the high dynamics range and non-saturation of the detector. Therefore, the accurate position and intensity of all spots can be extracted for the measurement of in-plane and out-of-plane structures (more details in Methods and Fig. S4 and S5).

Considering that FE SnSe has been predicted to have an in-plane lattice displacement between Sn and Se atoms oriented in either the [100] or [010] direction[18,22] (Fig. 1f)—lowering the symmetry to the $Pnm2_1$ space group (or non-centrosymmetric unit cell) from its parent paraelectric phase (Fig. S6)—the ferroelastic strain that coupled with FE polarization can be extracted from the positions of the spots in the diffraction patterns. To maximize this effect, we measured the position of {200} diffraction spots and calculated the reciprocal lattice vectors, which we have labeled as $\overline{AC}$ and $\overline{BD}$ in Fig. 1f. These vectors represent two potential elongation directions of the SnSe lattice, which are also called armchair directions due to armchair-shaped lattice (Fig. S6c). We mapped the ferroelastic strain (Fig. 1h, Fig. S1, Fig. S2), defined as $|\overline{AC}|/|\overline{BD}|-1$, and observed an intertwined domain morphology and a domain width that mostly ranged from 20 to 80 nm in thin-flake SnSe (Fig. S7). The observed ferroelastic strain (3.02%) was slightly larger than previous DFT simulations (2.1 %)[23,26], indicating a larger ferroelastic effect in thin SnSe flakes than the predicted monolayer system. In addition, by probing the rotation of the armchair direction in each domain (details in Methods), we observed an ~90° angle between parallel twin domains. Since the armchair direction in SnSe lattice is parallel to the electric polarization, the ~90° twin walls that we observed are consistent with previous works[17]. In addition, we observed a small lattice rotation (~5°) between red and blue (or yellow and purple) stripe domains (Fig. 1i, Fig. S1, Fig. S2). Such lattice rotation may lead to lattice distortions when these stripe domains meet perpendicularly, which can directly be seen in the strain and rotation maps at the superdomain boundaries (black boxes in Fig. 1h, i).

Besides the in-plane lattice structure, the out-of-plane stacking order affects the intensity of the diffraction spots (Fig. 1g). The simulation of diffraction patterns shows that the destructive interference at {110} Bragg peaks in "AC" stacking lattice leads to a much lower {110} spot intensity (Fig. 1g), while this effect does not appear in "AB" stacking case. Therefore, we utilized the integrated intensity of four {110} spots to indicate the presence of "AB" stacking in the SnSe thin flake. From our 4D data, we identified that most stripe domains contain a mixture of AC and AB stacking orders (Fig. 1j, Fig. S1, Fig. S2), which is consistent with existing publications[29]. However, we also observed the existence of almost pure "AC" stacking stripe domains, which form

twin walls with "AB" stacking stripes in the SnSe thin flake (Fig. 1j, yellow box). The formation of FE-AFE domain walls contributes to stabilizing FE domains, offering valuable insights for engineering anisotropic domain structures with exceptional non-linear optical responses. The structure and properties of such periodic FE-AFE domains will be discussed in greater detail later in the manuscript.

### Width-dependent deformation at 180° domain walls

Based on the in-plane mapping from our 4D data, we observed regular twin walls in vdW SnSe (Fig. 2a), which is consistent with the in-plane monolayer SnTe as measured by STM[17,36]. However, due to the rhombohedral lattice geometry, the [110] direction shows a deviation of ~5°, as captured in the diffraction patterns from our 4D data (Fig. 2b). The deviation leads to an 87.5° angle between the polarizations in neighboring stripe domains. Due to the non-90° twin walls between parallel stripes, the 180° domain walls where the stripe nanodomains meet perpendicularly must contain deformations to compensate for lattice mismatch (Fig. S8).

The measured ferroelastic strain (Fig. 2c) and rotation (Fig. 2d) maps indicate two types of morphologies: needle tips and "T" shaped junctions. Different from conventional superdomain boundaries in bulk ferroelectics[37,38], our maps suggested such deformations have a dependence on the in-plane width of these nano stripe domains. For example, when the horizontal stripe ($a_1^+$) that intersects with the vertical stripes is wider, a relatively larger in-plane continuous lattice rotation is introduced to the junction area (along the black arrow in Fig. 2d). The line profile of the rotations confirms a change of ~2° across 150 nm (Fig. 2e black), leading to the formation of the needle tips in $a_1^-$ domains. In contrast, a narrower horizontal stripe contains smaller continuous lattice rotation (blue arrow in Fig. 2d, Fig. 2e blue) but more lattice distortion at the intersection (black circles in Fig. 2c, d), forming "T" shaped junctions. The transition from needle tips to "T" junctions is also captured, showing a relatively smooth transition between two cases (dotted lines in Fig. 2c, d).

To quantify the in-plane width dependence of the domain morphology (needle tip or "T" junction), we analyzed ~50 stripe intersections in SnSe and summarized the geometries statistically. For simplicity, we refer to the width of the horizontal stripe at the boundary ($a_1^+$) as $b$ and the width of the vertical domain ($a_1^-$) as $a$. We measured the deformed length, $d$, of each vertical stripe at the superdomain boundary by the image contrast in Fig. 2d (details in Fig. S9). The deformed length is positive when $a_1^-$ deforms $a_1^+$ and generates "T" junctions (Fig. S9a, b), and negative when a gap is observed in between, which usually form needle tips (Fig. S9 c). A simple relationship is observed between the width ratio ($b/a$) and the normalized deformed length, which is defined as the ratio between deformed width and the original width of each $a_1^-$ domain ($d/a$) (Fig. 2f). We observed that T-junctions are favored when the width ratio is small. Conversely, as the width ratio increases, needle tips become more prevalent. A transition zone exists between these two configurations from 2.5 to 3 (Fig. 2f). This observation is consistent across multiple samples (Fig. S10), which indicates its ubiquity in vdW SnSe thin flakes.

To understand these two configurations, we performed a phase-field simulation to model the superdomain structures. Similar to experimental domain patterns, the in-plane domains are constructed in which the width ratio ($b/a$) between the horizontal and vertical domain stripes is continuously adjusted from 1:1 to 6:1. After a relaxation, the equilibrium domain structures show a transition from "T" junction structures in lower ratio cases to needle tips in higher ratios (Fig. S11a). Such effect is consistent with our experimental observations. To further understand the formation of the two configurations, we plot the elastic and electrostatic energy density distributions for each $b/a$ ratios (Fig. S11 b,c). It is clearly seen that the locally high elastic energy and the electrostatic energy density are

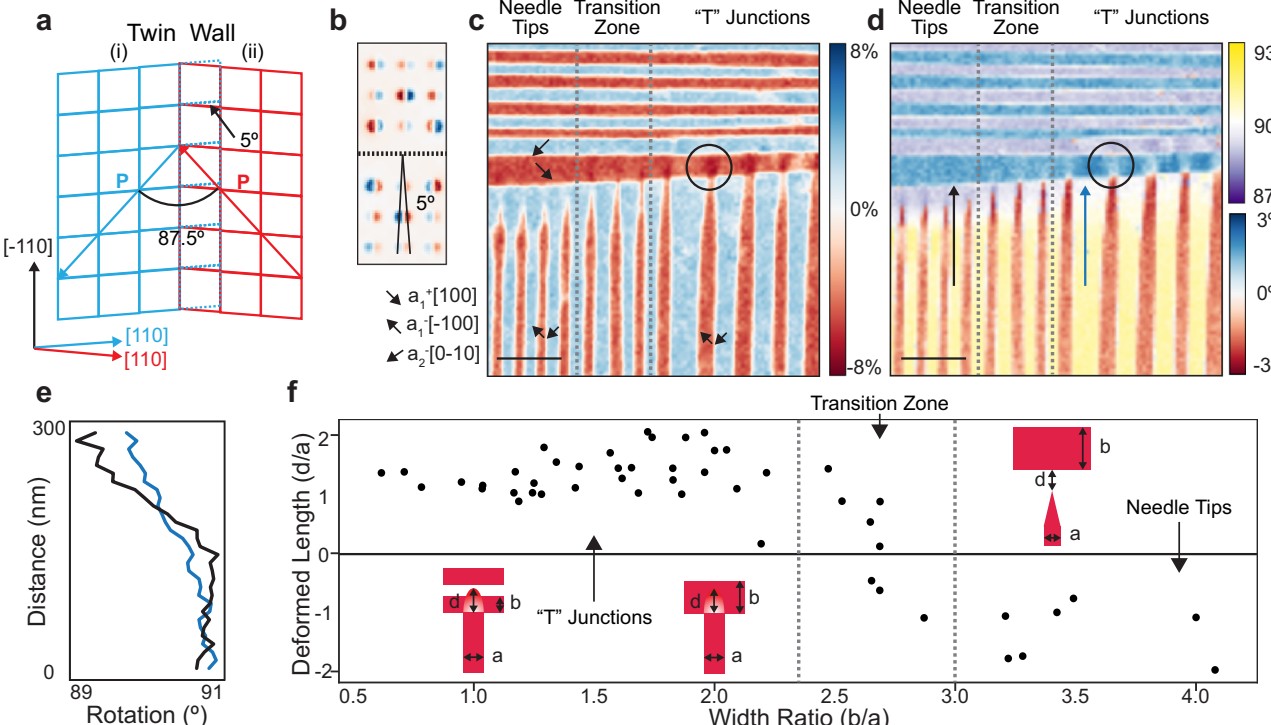

**Fig. 2 | Deformations at superdomain junctions. a** Lattice schematic of a twin wall. The twin domains are indicated in red and blue, respectively, with their polarization labeled by arrows. The original lattice without twinning is shown in dashed blue lines, illustrating a 5° mismatch of the [110] directions. **b** The difference of diffraction patterns from neighboring stripe domains, confirming the 5° rotation. **c, d** Ferroelastic strain (**c**) and rotation (**d**) maps at a superdomain boundary where both needle tip morphology and "T" junctions coexist. Polarization directions are indicated by arrows in (**c**), and a deformed "T" junction is circled. Between two dotted lines is a transition zone where both needle tips and "T" junctions are observed. The black and blue arrows in (**d**) highlight the differences in lattice rotation between needle tip structures and "T" junctions. (Scale bar: 200 nm) (**e**) Line profiles of the continuous lattice rotation from the black and blue arrows in (**d**) respectively. **f** Statistics of deformed length depending on the in-plane width ratio ($b/a$) of the stripe domains. The gray dotted lines show the transition zone. Inset: schematics of the measurements on widths ($a$ and $b$) and deformed lengths ($d$).

concentrated at the "T" junction boundary where $a_1^+$ and $a_1^-$ meet perpendicularly. This is because lattice distortion and bound charge are significant at the head-to-head 180° domain walls. For small domain width ratio cases ($b/a = 1:1$ or $2:1$), the local elastic energy in the horizontal $a_1^+$ domains (red boxes with blue regions in Fig. S11 b) is much lower than that along the interface. Therefore, the high elastic energy at the "T" junction superdomain boundary can potentially be released by propagating across the $a_1^+/a_2^-$ domain walls along [−110] direction, which further distorts the horizontal $a_1^+$ domain and forms the "T" junction structure. On the other hand, when the domain width ratio increases, the elastic energy in the wider horizontal $a_1^+$ domain also increases (red boxes with yellow regions in Fig. S11 b) and becomes close to that along the interface. In this case, the elastic energy relaxation towards the horizontal $a_1^+$ domain regions is partially inhibited. Consequently, the horizontal $a_1^+$ domain can hardly be deformed. Therefore, to reduce the elastic energy at the interface, a transition region of horizontal $a_2^-$ domain is formed between $a_1^+$ and $a_1^-$ domains to avoid the head-to-head 180° domain wall, so the vertical $a_1^-$ domains shrink and form needle tips.

## Built-in strain at "T" junctions

To further investigate how the lattice explicitly deforms, we further analyze individual localized regions from the 4D data at a "T" junction (boxes in Fig. 3a, b). The diffraction patterns from different areas (1-6) reveal that the "T" junction contains lattice rotations and local uniaxial strain (Fig. 3c, Fig. S12). To reveal the local lattice structure, we derived the deformation step by step from the rigid lattice model (Fig. 3d), which shows the 2.5° gaps due to the non-90° twin structure. The nanobeam diffractions indicate that the vertical stripes (box 3 and 4)

rotate ~1.3° clockwise (Fig. S12b, c), while different regions in the horizontal stripe rotate in the opposite direction (0.9° in box 2 and −0.5° in box 1) (Fig. S12d, e). This leads to an almost seamless boundary between 1 and 3 regions, as well as a ~ 3.3° lattice rotation between 2 and 4 (Fig. 3c, top). By adding the local rotation, the schematic shows distorted "T" junctions with the gap structure rearranged (Fig. 3e). However, lattice rotation is not sufficient to bridge the gaps. Our nanobeam diffractions reveal a large uniaxial strain (4%) in region 2 along the polarization direction (Fig. 3c bottom, Fig. S13 and S14). With strain added to the schematic illustration, the results show a lattice matched structure at the "T" junction without gaps or dislocations (Fig. 3f), mitigating crack initiation at the domain walls. We applied similar analysis to the needle tip structure (Fig. S15). Since the wider $a_1^+$ domain is harder to be deformed than in the "T" junction structure, the gap caused by the lattice rotation (~2.5°) needs to be compensated by the continuous rotation mechanism, which explains our observation of the lattice rotation spanning ~150 nm. Lastly, a large uniaxial strain is presented in $a_1^-$ domains to form a lattice matched structure. And to reduce the strain energy, the domain width shrinks and forms the needle tip[39].

Additionally, we investigated local areas where the "T" junctions have 1:1 and 2:1 width ratio ($b/a$). The rotation maps show that the superdomain boundary contains three "T" junctions with similar geometry for each case (Fig. 3g, k). We plotted the line profiles at each junction (following arrows in Fig. 3g,k) and removed the background rotation from neighboring areas (dotted lines in Fig. 3g,k). The averaged strain line profiles (Fig. 3h) show that the two cases adopt similar transitions at the junction, where the largest strain forms at the interface and drops when moving away from the boundary. In

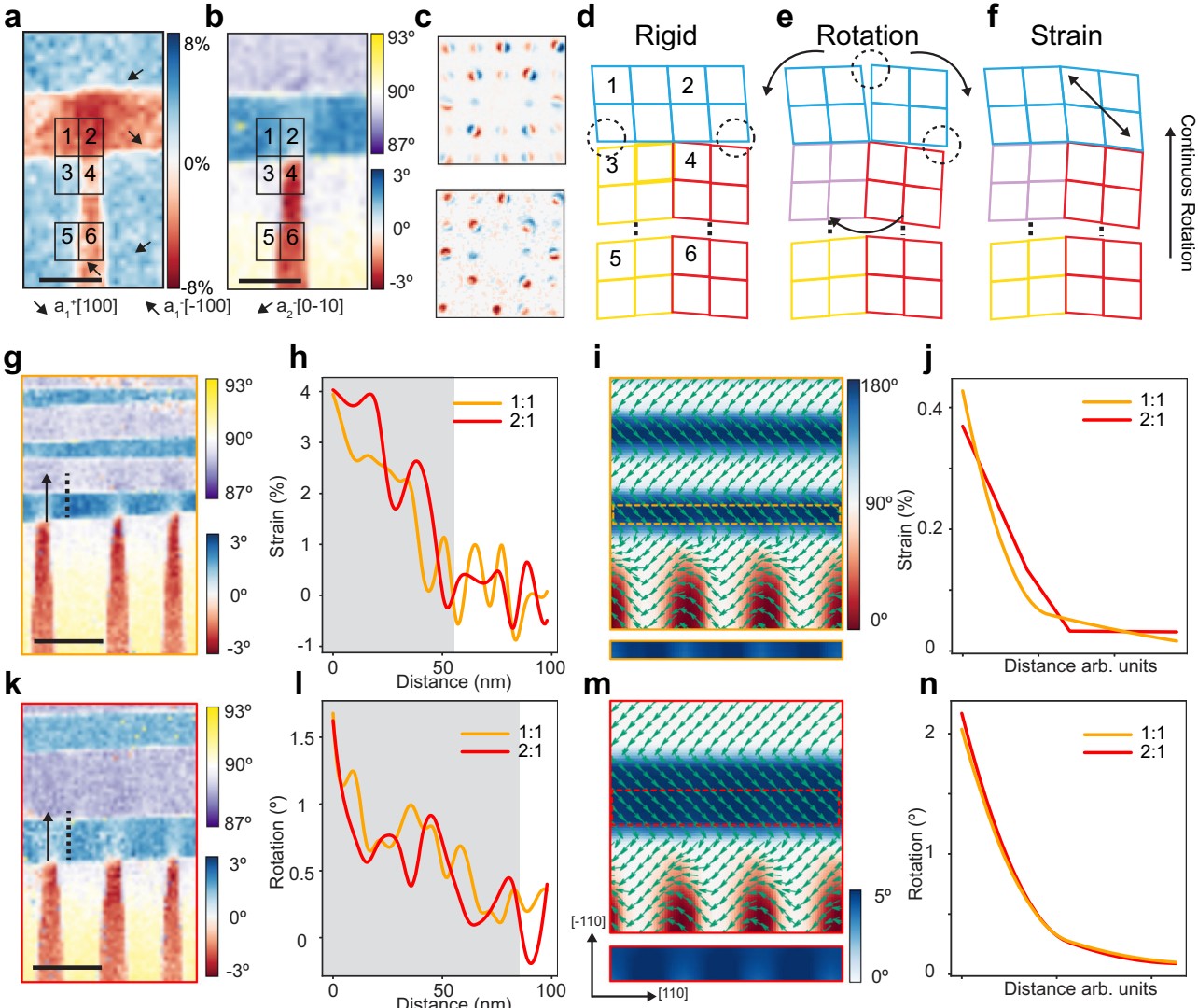

**Fig. 3 | Built-in strain gradient at T junctions. a, b** Zoomed-in maps of a "T" junction with six boxes labeled (the mean diffraction patterns from these six areas are shown in Fig. S12). (Scale bar: 50 nm) (**c**) Diffraction pattern differences between 2 and 4 (top) and 1 and 2 (bottom), indicating 3.3° lattice rotation and 4.0% uniaxial strain, respectively. **d–f** Real space lattice schematic of the "T" junctions for the rigid model (**f**), model that contains lattice rotation (**e**), and model that contains both rotation and strain (**f**). Gaps are labeled in dashed circles. The colors match those in the rotation map (**b**). **g–k** Rotation maps of the "T" junctions with width ratio 1:1 (**g**) and 2:1 (**k**). (Scale bar: 100 nm) (**h, l**) Line profiles of the built-in strain (**h**) and rotation (**l**) from the 1:1 (yellow) and 2:1 (red) "T" junctions. Gray shades indicate the gradient of strain and rotation. **i–m** Phase field simulation of the rotation maps from T junctions with ratio 1:1 (**i**) and 2:1 (**m**), with the polarization vector (green arrows) overlaid on the images. The first horizontal domains (yellow and red dashed boxes) are displayed below with an adjusted color scale to show the deformation contrast (0° to 5°). **j–n** Line profiles of the build-in strain (**j**) and rotation (**n**) from the simulated "T" junctions.

particular, for 1:1 ratio "T" junctions, the deformation can penetrate the $a_1^+/a_2^-$ domain walls and affect the top $a_2^-$ domains (Fig. 3g). The line profile also indicates a built-in strain gradient across ~50 nm (gray area in Fig. 3h). Similarly, lattice rotation also decreases as distance from the interface increases, with a smoother gradient over ~80 nm (Fig. 3l). The results show no abrupt lattice changes across horizontal twin walls, which avoids significant lattice distortions that initiate defects and cracks.

Our phase field simulation results showed similar effects, where the horizontal domains ($a_1^+$) with width ratios ($b/a$) 1:1 or 2:1 are deformed periodically by the vertical stripes ($a_1^-$), as shown in Fig. 3i and m. The largest lattice rotation and deformation occur at the $a_1^+/a_1^-$ domain interface, and decay along the [−110] direction, inducing gradients in both built-in strain and lattice rotation (Fig. 3j and n). That is because the high elastic energy at the "T" junction superdomain boundary tends to be released by propagating across the horizontal $a_1^+/a_1^-$ domain walls, causing a strain gradient at this region. In contrast,

the head-to-tail $a_1^+/a_2^-$ 90° domain walls in the horizontal domain regions are both elastically coherent[40] and charge neutral, which exhibit lower elastic and electrostatic energy density (Fig. S11). Thus, these low energy $a_1^+/a_2^-$ domain walls have limited effects on the lattice rotation and strain gradience induced by the high elastic energy from $a_1^+/a_1^-$ domain walls. Therefore, the simulated deformed lengths are almost identical under different domain width ratios ($b/a$), which agrees with the trend in experimental observations.

**Ferroelectric to antiferroelectric twin walls**

In addition to in-plane structures, out-of-plane stacking order also affects domain wall properties. To map the out-of-plane stacking order using an in-plane scanning electron beam, we probed the intensity of {110} diffraction spots, which are forbidden in AC (or FE) stacking but allowed in AB (or AFE) stacking (Fig. 1g). Since our SnSe flake has a thickness of ~12 nm[33] (~20 layers with 10 unit cells), it may contain a mixture of AB and AC stacking. To quantitatively determine the ratio

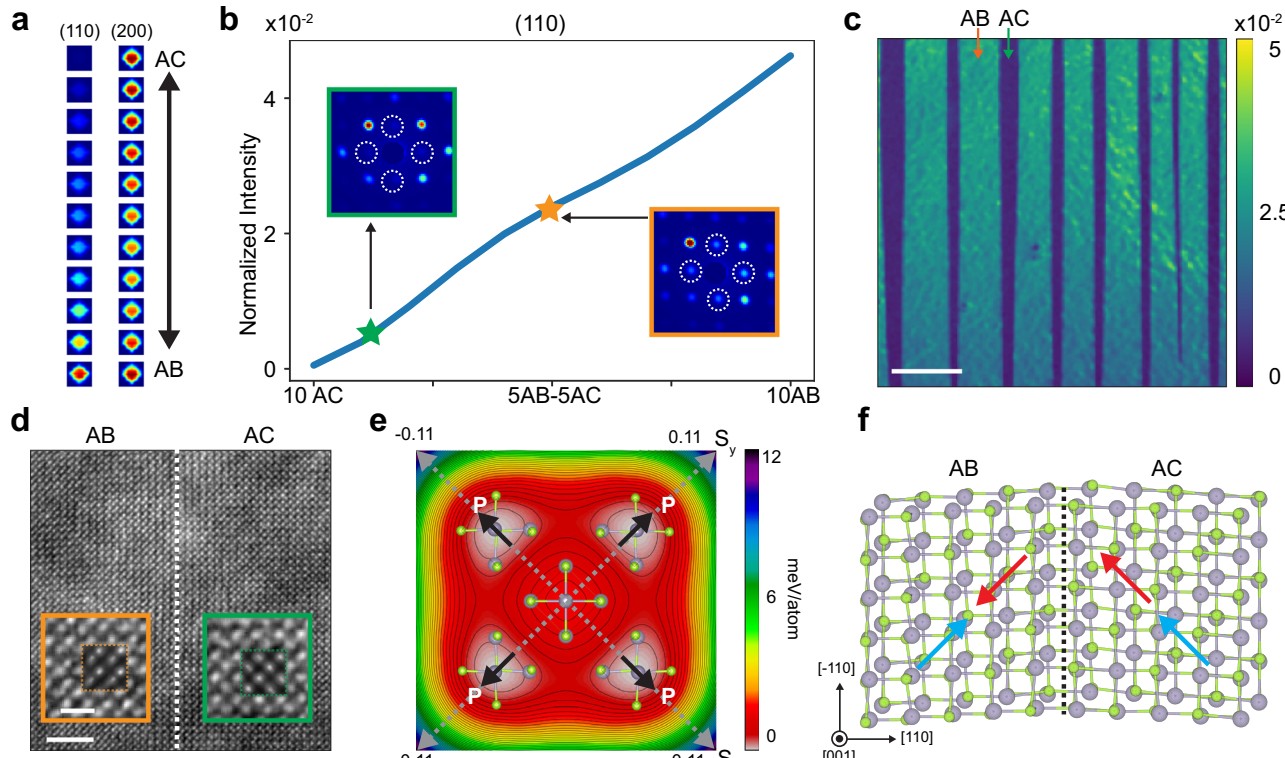

**Fig. 4 | Stacking orders in SnSe. a** Simulated intensity of {110} and {200} diffraction spots for different stacking orders. **b** Plot of normalized intensity ratio (details in Methods), which changes as the stacking order varies in our simulated lattice model. The plot is annotated with experimental measurements from (**c**) with two representative experimental diffraction patterns of AC-dominate (FE) domain (green star) and AB-dominate (AFE) domain (orange star) in the insets. **c** Normalized intensity ratio map from region contains both FE and AFE domains, forming periodic FE-AFE domain walls. (Scale bar: 500 nm) The averaged diffraction patterns from FE rich (green arrow) and AFE rich (orange arrow) domains are shown in the insets of (**b**) with the quantitative stacking information labeled as star shapes. The intensity variance in the AFE rich domains is from the slightly lattice tilt (Fig. S26). **d** Atomic resolution ADF-STEM image of one FE-AFE domain wall. (Scale bar: 2 nm) Insets: Zoomed-in ADF-STEM images overlayed with simulations showing the AFE "dumbbell" lattices and FE square lattices. (Scale bar: 0.5 nm) **e** DFT-calculated of the ferroelastic and ferroelectric potential energy surface with atomic structures superimposed at local minima and unstrained maxima. **f** Relaxed atomistic structure of FE-AFE twin boundary in bilayer SnSe. The twin boundary is indicated by a dashed line. Red and blue arrows indicate the electric polarization direction in the $x$–$y$ plane for top and bottom layer, respectively.

between AC and AB stacking, we simulated diffraction patterns of 10-unit-cell thick SnSe lattices that contain different ratios of AC and AB stacking orders using the kinetic diffraction theory model[41] (Fig. S16 and S17). As the AB stacking order in the flake increases, there is a gradual increase in the intensity of the {110} spots, while the intensity of the {200} spots remains relatively unchanged (Fig. 4a). The simulated data shows an approximately linear relationship between the intensity of {110} spots and the stacking ratio (AB/AC) (Fig. 4b). We utilized the normalized intensity ({110} intensity /total intensity of the diffraction pattern) for comparison with experimental results (details in Methods). Figure 4c shows the normalized intensity map of the FE-AFE domain walls from a 4D data. The simultaneous strain and rotation maps (Fig. S18) indicate the FE-AFE domain walls are located at the 90° twin walls. The histogram (Fig. S19) of the normalized intensity map (Fig. 4c) shows two peaks at 0.54 and 2.5 (x10$^{-2}$), indicating a ~90% AC stacking in FE stripe domains and a ~50% AB stacking in AFE domains, respectively (Fig. 4b). To further validate our method, we acquired the atomic-resolution ADF-STEM image at the FE-AFE domain wall (Fig. 4d). The zoomed-in images reveal the "dumbbell" lattice (left) and square lattice (right) from neighboring stripe domains (schematics shown in Fig. S20), indicating that the two neighboring domains adopt different stacking sequences and form FE-AFE domain walls.

To understand the formation of FE-AFE domain walls, we performed first-principal calculations to simulate the ferroelastic and ferroelectric potential energy of SnSe (Fig. 4e), which shows that individual SnSe layer possesses four degenerate in-plane

ferroelectric-ferroelastic variants and each variant can transform into two adjacent variants along the minimum energy pathway via in-plane Sn-Se local dimer rotation of ~90° (Fig. S21). This explains the stable ~90° twin walls in individual SnSe layer. We further simulated the ~90° twin walls with head-to-tail configuration and tail-to-tail configuration (Fig. S22). Different from the head-to-tail configuration (Fig. S23), we discovered that the tail-to-tail structure relaxes to a different phase due to its local instability. We further calculated a bilayer twin wall with AFE (AB) stacking and FE (AC) stacking (Fig. 4f). In contrast to observations from monolayer twin walls, the FE-AFE domain wall relaxed to a locally stable structure without broken bonds. We note that due to the stacking order changes, the FE-AFE twin walls include either head-to-head or tail-to-tail twin walls (Fig. S24), which are unstable in their monolayer form. However, due to the vdW force, the interlayer interaction stabilizes these twin walls and allow the formation of FE-AFE domain walls. In addition, the DFT calculated structure of the neighboring FE and AFE domains agrees better with our experimental measurements than the monolayer model and other reported structures (Table S1). The results explained the existence of meta stable FE-AFE domain walls, which is expected to have anisotropic and strong nonlinear optical responses such as second harmonic generation (SHG) and linear electro-optic effects[20,21]. We conducted SHG measurements on SnSe flakes and observed angle-dependent polarization across different regions, confirming the influence of stacking on nonlinear optical responses of these flakes (Fig. S25).

## Discussion

In conclusion, our study provides an in-depth understanding of domain-dependent in-plane strain and out-of-plane stacking effects in single-crystalline vdW FE thin flakes across a multi-micrometer scale. Our approach surpasses the constraints of traditional imaging techniques, facilitating the simultaneous mapping of in-plane and out-of-plane structure across the entire flakes. The "T" shaped junction and needle-tip morphology we identified in vdW FEs have profound implications, especially in influencing switching energy and their viability in memory and actuator device applications. Our insights into domain width dependence pave the way for strain control through domain size engineering. The identification of the coexisting FE and AFE domains unveils FE-AFE domain wall structures, presenting a fresh avenue to fine-tune electronic and optical attributes. Notably, the deformed T-junctions and FE-AFE twin walls observed in this study are unprecedented in two-dimensional FEs and distinguishable from conventional bulk and epitaxial thin film FEs. Our work paves the way for domain engineering in 2D vdW FEs, which could serve as building blocks for future device applications.

## Methods

### Material synthesis

2D SnSe crystals were synthesized on mica substrates by low-pressure PVD using commercially available SnSe powder (99.999% (metals basis), Thermo Scientific Chemicals) as the precursor. A quartz boat filled with SnSe powder was placed in the center of a single-zone tube furnace, and a piece of mica was placed downstream (~10 cm from the SnSe precursor) as the growth substrate. During the synthesis process, the PVD system was first pumped down to a base pressure of ~10 mTorr. Subsequently, the furnace was ramped up to 440 °C in 10 min and held for 45 min for SnSe growth. Afterwards, the furnace was opened for rapid cooling. A mixture of 65 sccm Ar and 5 sccm H2 was used as the carrier gas throughout the synthesis process. SnSe sample was then transferred onto TEM grid using PMMA as the supporting layer and sonication to release the sample from the original substrate. SHG imaging was conducted to select targeting flakes with ferroelectric domains.

### EMPAD data acquisition

The 4D-STEM datasets were taken on an aberration-corrected FEI Titan Themis with an Electron Microscope Pixel Array Detector (EMPAD)[33]. The 4D datasets of the SnSe thin flake were acquired at 300 kV. A 0.5-mrad convergence angle was used, leading to a ~2.44 nm probe size (defined by Full Width Half Maximum (FWHM) probe diameter) and 4.88 μm depth of focus[42] (Fig. S27). For a 300 kV electron beam, 579 ADUs represent one electron per pixel. For all the datasets, an exposure time of 1.86 ms (1 ms acquisition time along with 0.86 ms readout time) was employed when acquiring the EMPAD 4D datasets. The scan size in real space (the number of pixels the beam scans across) can be set from 64 × 64 to 512 × 512. The scan size of the data used in this work was 256 × 256. The total time for capturing one 4D data was 122 s. In addition, the data in Fig. 1 and Fig. 2 were taken at 37k magnification with a 2.6 μm field of view (FOV), and the data in Fig. 3 and Fig. 4 were taken at 75k magnification with a 1.3 μm FOV. A camera length of 720 mm was employed to ensure clear separation between the {110} spots and the {200} spots. This distinction allows our strain mapping (based on the {200} spots) and stacking mapping (based on the {110} spots) to be entirely independent.

### 4D data processing

The determination of each diffraction spot's location relies on its Center of Mass (CoM). To generate initial masks, we compute the mean diffraction patterns across the entire dataset (Fig. S4a) and subsequently select the four {200} diffraction spots to create circle masks

(Fig. S4b). The centers of these circle masks are determined as the pixels with the highest intensity within their respective diffraction spots. Following this, we calculate the CoM using the initial circular mask (indicated by the white dotted circle in Fig. S4c). However, due to variations in strain and rotation from different scan positions, the centers of the diffraction spots may deviate from the mask center determined by maximum intensity. This deviation can lead to minor errors due to the unbalanced background contribution in the CoM calculation. To resolve this issue, we shift our masks to center them around the CoM calculated from *each* diffraction pattern (as shown by the red dotted circle in Fig. S4c). This correction utilizes adaptive masks and is performed iteratively until convergence is achieved, which we define as the difference between two iterations falling below 0.01 pixels (Fig. S4d). A comparison is made between the fixed mask and the adaptive mask methods (Fig. S5), demonstrating that the latter approach effectively corrects drift errors and provides more accurate center measurements. With the converged CoM values for the diffraction spots from the adaptive mask approach, we proceed to compute the length and orientation of the reciprocal vectors AC and BD in Fig. 1f.

The ferroelastic strain maps and the rotation maps are generated based on the reciprocal vectors. Following the definition of the ferroelastic strain[23], we calculated it based on the equation $|\overline{AC}|/|\overline{BD}|-1$. The ferroelastic strain represents the lattice constant differences between the armchair and zigzag directions (Fig. S6c, d), where electric polarization is along the armchair direction. The rotation map is calculated by measuring the angle of the armchair direction. In the reciprocal lattice, $\overline{AC}$ or $\overline{BD}$ with the smaller length represents the armchair direction. And we utilized the averaged angle of the $\overline{AC}$ vector as the zero. Therefore, the rotation angle for different areas is calculated by measuring the angle between the armchair direction and the averaged $\overline{AC}$. Another way to express this is $<\text{mean}(\overline{AC}), \min(|\overline{AC}|,|\overline{BD}|)>$, where mean($\overline{AC}$) is the averaged vector of $\overline{AC}$, min($|\overline{AC}|,|\overline{BD}|$.) is the vector with smaller magnitude, whether it is $\overline{AC}$ or $\overline{BD}$, and the angle brackets indicate the angle measurement. For example, if $\overline{AC}$ has the elongated lattice, the rotation angle varies from −3° to 3°, shown as red and blue domains in the rotation map (Fig. 1i). If $\overline{BD}$ is along the armchair direction, the lattice rotation varies between 87° and 93°, shown as yellow and purple in the rotation map (Fig. 1i). We devised this mapping strategy in response to the limitations of conventional measurements from the strain matrix, which often fail to clearly delineate polarizations and their angles in complex, intertwined domains. In the conventional method[35], the four strain maps ($e_{xx}$, $e_{yy}$, shear, and rotation) are derived based on a selected basis (Fig. S28). While it's theoretically possible to compute the ferroelastic strain and armchair rotation from the shear and rotation maps produced by the conventional method (see Fig. S29), the presence of deformations at superdomain boundaries, such as the continuous lattice rotation, complicates the decoupling of shear and rotation, often yielding imprecise results. In contrast, our method directly assesses the ferroelastic strain and rotation from diffraction patterns, producing more accurate maps.

In order to generate the {110} intensity maps, we first generated circular masks on the four {110} spots from linear combination of {200} reciprocal vectors. We then determined the {110} intensity by integrating the intensity within the {110} circular masks. To account for the intensity variation of {110} spots in AFE domains due to the sequence (Fig. S17) and minor lattice tilts (Fig. S26), we summed the intensity of all four {110} spots to map the stacking order in our SnSe thin flakes. To quantify the stacking order and provide a comparison consistent to simulations, we normalized the summed intensity of the {110} spots by dividing it by the integrated intensity of the entire diffraction pattern. We opted not to utilize the {200} spot intensity for normalization since it is more sensitive to variations caused by lattice tilts.

## First-principles DFT calculations

Atomistic and electronic structures of monolayer SnSe were calculated using first-principles density functional theory (DFT)[43,44] using the Vienna Ab initio Simulation Package (VASP)[45] and projector augmented wave method for treating core electrons[46]. We used the Perdew-Burke-Ernzerhof (PBE) form of exchange-correlation functional within the generalized gradient approximation (GGA)[47], plane-wave basis with a cutoff energy of 450 eV, and a Monkhorst-Pack[48] k-point sampling grid of $8 \times 8 \times 1$. Ground state structures of monolayer SnSe were obtained by fully relaxing both atomic positions and in-plane lattice parameters while keeping a vacuum region of 12 Å along the plane normal (i.e. the z axis) to avoid the periodic image interactions. The maximal residual atomic force for structural relaxation was set to be 0.02 eV Å$^{-1}$, and the convergence criteria for electronic relaxation is set to be <10$^{-6}$ eV. To understand ferroelectric and ferroelastic transitions, we carried out potential energy surface calculations by constraining the relative fractional $x$–$y$ coordinates of Sn-Se pairs and the lattice parameter along z, and relaxing both cell parameters and shape in the $x$–$y$ plane as well as cartesian coordinates of all atoms along z direction. For accurate potential energy surface calculation, we used a higher plane-wave cutoff energy of 520 eV, a denser k-point sampling grid of $12 \times 12 \times 1$, a vacuum region of 15 Å, the maximal residual atomic force of 0.01 eV Å$^{-1}$, and the convergence criteria for the electronic relaxation of <10$^{-7}$ eV.

## Phase field method

In the phase-field simulations, we use polarization vector $\mathbf{P_i}$ as the order parameters to describe the ferroelectric state. The equilibrium domain structure is determined by minimizing the total free energy ($F$) with respect to $\mathbf{P_i}$ via solving the time-dependent Landau–Ginzburg–Devonshire (LGD) equations,

$$\frac{\partial P_i(r,t)}{\partial t} = -L\frac{\delta F}{\delta P_i(r,t)}, (i=1,2,3) \quad (1)$$

where $L$ represents the kinetic coefficient related to the domain wall mobility, $t$ is time, $r$ is the spatial position. The total free energy $F$ includes the Landau, gradient, elastic, and electrostatic energies,

$$F = \int_V (f_{land}(P_i) + f_{grad}(P_{i,j}) + f_{electric}(P_i, E_i) + f_{elastic}(P_i, \varepsilon_{ij})) dV \quad (2)$$

where V is the total volume of the system, and $\varepsilon_{ij}$ and $\mathbf{E_i}$ represent the components of strain and electric fields. Since the magnitude and in-plane orientation of spontaneous polarizations in 2D SnSe ferroelectric materials and PbTiO$_3$ are similar[18,49], in this simulation, we use a six-order Landau polynomial function for SnSe 2D ferroelectrics, and select the corresponding Landau energy coefficients to be same as in PbTiO$_3$. Detailed expressions of each free energy density in Eq. (2) can be found in the Ref. 49. The simulation system is chosen to be $100\Delta x \times 100\Delta y \times 32\Delta z$, with $\Delta x = \Delta y = \Delta z = 1.0$ nm. The thickness of the film, substrate, and air are $20\Delta z$, $10\Delta z$, and $2\Delta z$, respectively. The temperature is $T = 25\,°C$, and an isotropic relative dielectric constant ($\kappa_{ii}$) is chosen to be 50. To mimic the 2D in-plane ferroelectric domain structure, the substrate strain is set to be 5% to keep the global a$_1$/a$_2$ structure stable. The Landau coefficients, electrostrictive coefficients, and elastic-compliance constants are collected from Ref. 49.

## ADF-STEM imaging and simulation

ADF-STEM image is taken in 300 kV, and the dwell time is 10 μs at each scan position. The convergence angle of the electron probe is 25 mrad, with a 115 mm camera length. The simulated ADF-STEM images are generated by the multi-slice method in abTEM[41] package with the same conditions.

## Data availability

The 4D-STEM datasets generated in this study have been deposited in the Zenodo database https://zenodo.org/communities/hanlab-rice/.

## Code availability

The data processing code can be accessed in this GitHub link below: https://github.com/Chuqiao2333/2D_ferroelectric_SnSe/.

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

## Acknowledgements

The work is supported by National Science Foundation (NSF) under the award number CMMI-2239545. C.S. and Y.H. acknowledge the support from Welch Foundation (C-2065). N.M., T.Z. and J.K. acknowledge the support by the U.S. Department of Energy (DOE), Office of Science, Basic Energy Sciences (BES) under award DE-SC0020042 and the support from the STC Center for Integrated Quantum Materials, NSF grant number DMR-1231319. K.Z. and Y.C. and acknowledges the support from the NSF under the award number NSF 2132105. K.A. and X.Q. acknowledges the support from the Center fro Reconfigurable Electronic Materials Inspired by Nonlinear Dynamics (reMIND), an Energy Frontier Research Center funded by DOE BES under award DE-SC0023353. X.T. acknowledges support from NSF under DMR-1753054. Portions of this research were conducted with the advanced computing resources provided by Texas A&M High Performance Research Computing. This work made use of the electron microscopy facility of the Platform for the Accelerated Realization, Analysis, and Discovery of Interface Materials (PARADIM), which is supported by NSF under Cooperative Agreement No. DMR-2039380. This work made use of the Cornell Center for Materials Research Shared Facilities which are supported through the NSF MRSEC program (DMR-1719875). The authors thank Guanhui Gao and Hua Guo for maintaining our electron microscopes. The authors thank Dr. William A. Tisdale at MIT for assistance of SHG measurement. The phase-field simulation results were obtained using the software package Mu-PRO (www.mupro.co).

## Author contributions

C.S. and Y.H. conceived the project; 4D-STEM and data analysis were carried out by C.S. under the supervision of Y.H.. Diffraction simulation was conducted by C.S. and G.G. under the supervision of Y.H.. Sample growth was done by T.Z. and M.-H.C. under the supervision of J.K.. N.M. transferred the sample and performed the SHG and PFM measurements. K.Z. conducted phase field simulation under the supervision of Y.C.. K.A. and X.T. performed DFT simulation under the supervision of X.Q.. B.W., L.C., and S.L. provided valuable suggestions. C.S. and Y.H. wrote the paper.

## Competing interests

The authors declare no competing interests.
