## [Peer Review File · Nature Communications]

Domain-dependent Strain and Stacking in Two-dimensional van der Waals FerroelectricsREVIEWER COMMENTS

Reviewer #1 (Remarks to the Author):

In this paper, the authors used a 4D-STEM technique to simultaneously probe in-plan strain and out-of-plane stacking in SnSe. They observed large lattice strain and the coexistence of ferroelectric and antiferroelectric stacking. These findings seem to be interesting, but their significance is not very clear, which requires to be further explored. In addition, I also have some questions about the research content of the article, seen below.

1. Is CoM a standard measurement of diffraction patterns? The author mentioned the condition of iterative convergence, but the iterative algorithm between the current and previous CoM was not discussed. I noticed that the 4D data processing was proposed by the authors, is it an algorithm that has been utilized by other researchers? In addition, the legend in 4D data processing is not correct, it is from Figure S1.
2. The probe size is ~3 nm, how long does it usually take to capture a 4D-STEM dataset, whose size is in the micron scale as shown in Figure 1. How do the authors adjust the focus of the probe size during 4D data processing, will the difference in focus at different positions affect the experimental results? How to judge and correct the scanning distortion? Will the surface fluctuation of TEM grid affect the experimental results? What is the detection thickness limit of 4D data processing?
3. The author claimed a large uniaxial strain (4%) in region 2, are there any other experimental evidences, such as HAADF images?
4. Even though locating in the "T" shaped junction area (Figure 2c), the second vertical stripe from right to left has a narrow width at the intersection, which is very similar to the needle tip.
5. The deformed length, d , is obtained by measuring the distance between the horizontal boundary and the tip of the vertical stripe according to the schematics of Fig. 2f. How is the deformed length actually measured, according to the diffraction spots or the image contrast in Figure 2c-d?
6. The author claimed that the critical width ratio is around 2.7. What's the physical mechanism of this value? Can the authors discuss further with the relation to the in-plane strain? Did authors conduct phase field in this state?
7. What is the distribution law of horizontal and vertical stripes width? Can the width of stripes be actively controlled? For example, changing the annealing time, temperature?
8. The author analyzed two types of the morphologies: needle tips and "T" shaped junctions. In the practical devices, what are the specific application advantages of the two superdomain boundaries?
9. The authors simulated diffraction patterns of 10-unit-cell thick SnSe lattices that contain different ratios of AC and AB stacking orders using the kinetic diffraction theory model in Figure S8. Why the intensity of two spots at the lower right corner changes more obviously than the two spots at the upper left corner?
10. The authors utilized the atomic model and diffraction spots of FE structure (AC) to investigate the in-plane strain in Figure 1-3. But the intensity of (110) spots ($\sim 1.4 \times 10^4$) in the majority area is obvious, corresponding to the AFE (AB) stacking ordering. Will this fact affect the in-plane strain results?
11. The authors utilized the normalized intensity ($\{\{110\}\}$ intensity / total intensity of the diffraction pattern) to investigate the stacking sequence variation. What does the "total intensity" refer to?
12. The authors mentioned that Figure S10 is the normalized intensity, but it is clear that the range of Y axis is much larger than 1. Also, what does the X axis stand for?
13. What's the parameter of the ADF-STEM image in Figure 4d? What's the simulation

condition for the inset simulated images?

Reviewer #2 (Remarks to the Author):

In this manuscript, by utilizing 4D-STEM, the authors probed the microscopic strain and rotation in 2D SnSe to visualize the complicated crystal structures around the domain walls. By this means, the authors could directly quantify the deformation and rotation of the crystals. They found strain as large as 4% which could exist in a 50 nm narrow domain wall region and a width-dependent T junction at the 180° domain walls. Meanwhile, the authors also verified the unique ferroelectric-antiferroelectric domain structures arising from different out-of-plane stacking. Overall, this research is of current interest and the manuscript is well organized. However, a few critical questions should be addressed before any recommendation can be given.

1. Through the manuscript, the authors only characterized one sample and even one region on the sample? The reviewer understands that the measurements might be challenging; however, is this sufficient to draw a general conclusion?
2. May be related to the comment 1, as the authors only characterized one sample, will the method and/or process of the preparation affect the domain structures? For example, since the AB and AC stacks in SnSe nanoflake represent different phases, would the mixed FE-AFE orders in SnSe nanoflake form due to the specific method and/or process?
3. Basically, the as-prepared 3D ferroelectric materials always contain various domain structures. What is the main feature of the domain structures probed in this manuscript. Is it possible to tune the domain structures?
4. Generally, the in-plane deformation includes rotation, normal strain and shear strain. Could the definition of strain and rotation in this work effectively differentiate these contributions? For example, will the shear strain also contribute to the rotation as defined?
5. Although a few studies have reported the ferroelectric properties of 2D SnSe, were it better if the authors also characterize, at least briefly, ferroelectric properties of their flakes. After all, ferroelectric properties depend on thickness, stacking and so on... In particular, the ferroelectric and antiferroelectric regions are claimed to coexist.

Point to Point response to reviewers

Reply from the authors in blue. Changes in the manuscript in red.

Reviewer #1 (Remarks to the Author):

In this paper, the authors used a 4D-STEM technique to simultaneously probe in-plan strain and out-of-plane stacking in SnSe. They observed large lattice strain and the coexistence of ferroelectric and antiferroelectric stacking. These findings seem to be interesting, but their significance is not very clear, which requires to be further explored.

We thank the reviewer for finding our work interesting. We apologize that our original manuscript might not have sufficiently emphasized our significance. After careful revision of our manuscript, we would like to point out our revised title “**Domain-dependent Strain and Stacking in Two-Dimensional van der Waals Ferroelectrics**”. This change emphasizes our contribution in elucidating the strain and stacking effects throughout vdW FE flakes and in unveiling novel domain-dependent morphologies and domain wall structures.

In addition, we have made modifications to our Conclusion section (Page 9, the second paragraph) to further highlight our study’s significance. “In conclusion, our study provides an in-depth understanding of **domain-dependent in-plane strain and out-of-plane stacking effects in single-crystalline vdW FE thin flakes across a multi-micrometer scale. Our approach surpasses the constraints of traditional imaging techniques, facilitating the simultaneous mapping of in-plane and out-of-plane structure across the entire flakes. The "T" shaped junction and needle-tip morphology we identified in vdW FEs have profound implications, especially in influencing switching energy and their viability in memory and actuator device applications. Our insights into domain width dependence pave the way for strain control through domain size engineering. The identification of the coexisting FE and AFE domains unveils FE-AFE domain wall structures, presenting a fresh avenue to fine-tune electronic and optical attributes. Notably, the deformed T-junctions and FE-AFE twin walls observed in this study are unprecedented in two-dimensional FEs and distinguishable from conventional bulk and epitaxial thin film FEs. Our work paves the way for domain engineering in 2D vdW FEs, which could serve as building blocks for future device applications.**”

1. Is CoM a standard measurement of diffraction patterns? The author mentioned the condition of iterative convergence, but the iterative algorithm between the current and previous CoM was not discussed. I noticed

that the 4D data processing was proposed by the authors, is it an algorithm that has been utilized by other researchers? In addition, the legend in 4D data processing is not correct, it is from Figure S1.

CoM (Center of Mass) is a mathematical definition and initially introduced to analyze diffraction patterns captured by the EMPAD detector in our previous work¹. In this manuscript, we utilized the CoM method with a more advanced iterative algorithm specifically designed for this purpose. We apologize for the lack of comprehensive algorithmic information in our previous manuscript. To address this concern, we now have included additional paragraphs in our methodology section, as well as two SI figures, elucidating the algorithm and highlighting the advancements of our current approach over previous methods.

We revised the following paragraph in our Method (Page 12, the third paragraph):

“4D Data Processing: The determination of each diffraction spot's location relies on its Center of Mass (CoM). To generate initial masks, we compute the mean diffraction patterns across the entire dataset (**Fig. S4a**) and subsequently select the four {200} diffraction spots to create circle masks (**Fig. S4b**). The centers of these circle masks are determined as the pixels with the highest intensity within their respective diffraction spots. Following this, we calculate the CoM using the initial circular mask (indicated by the white dotted circle in **Fig. S4c**). However, due to variations in strain and rotation from different scan positions, the centers of the diffraction spots may deviate from the mask center determined by maximum intensity. This deviation can lead to minor errors due to the unbalanced background contribution in the CoM calculation. To resolve this issue, we shift our masks to center them around the CoM calculated from *each* diffraction pattern (as shown by the red dotted circle in **Fig. S4c**). This correction utilizes adaptive masks and is performed iteratively until convergence is achieved, which we define as the difference between two iterations falling below 0.01 pixels (**Fig. S4d**). A comparison is made between the fixed mask and the adaptive mask methods (**Fig. S5**), demonstrating that the latter approach effectively corrects drift errors and provides more accurate center measurements. With the converged CoM values for the diffraction spots from the adaptive mask approach, we proceed to compute the length and orientation of the reciprocal vectors AC and BD in **Figure 1f**.”

We added the following SI figure in our Fig. S4:

Figure S4 (d) The workflow of the iterative CoM method to create adaptive masks, ensuring precise pinpointing diffraction spot positions. Upon establishing the initial mask (white dotted circle), the CoM within the mask is calculated as CoM₁. A subsequent circle mask (red dotted circle) centered on CoM₁ is applied (red). The CoM calculated using the red mask is determined as CoM₂. If the difference between CoM₂ and CoM₁ exceeds a 0.01 pixel threshold, the workflow is iterated until the difference is within the set threshold.

We added the following SI figure in our Fig. S5:

Figure S5 | Comparison of fixed-mask and the adaptive-mask methods for CoM measurements. (a) Four sample diffraction spots with the fixed masks, showing variations in different diffraction patterns. CoM values derived from these patterns are displayed below (in pixels). **(b)** Four sample diffraction spots with the adaptive masks, demonstrating evident improvements in mask alignment. A comparison of the calculated CoM values reveals that the adaptive-mask approach reduces the alignment inaccuracies caused by uneven backgrounds by up to an estimated 2%.

We acknowledge the reviewer for pointing out the error in indexing Fig. S1. We have carefully revised the manuscript and ensured that all SI figures are correctly referenced.

2. The probe size is ~3 nm, how long does it usually take to capture a 4D-STEM datasets, whose size is in the micron scale as shown in Figure 1. How the authors adjust the focus of the probe size during 4D data processing, will the difference in focus at different positions affect the experimental results? How to judge and correct the scanning distortion? Will the surface fluctuation of TEM grid affect the experimental results? What is the detection thickness limit of 4D data processing?

We thank the reviewer for the questions. We will address them point to point below:

(1) The probe size is ~3 nm, how long does it usually take to capture a 4D-STEM datasets, whose size is in the micron scale as shown in Figure 1.

For all the datasets, an exposure time of 1.86 ms (1 ms acquisition time along with 0.86 ms readout time) was employed when acquiring the EMPAD 4D datasets. The scan size of the data used in this work was 256×256. So, in total, it takes 122 seconds or around 2 minutes to capture one 4D data.

We add the following sentence in our Method of EMPAD data acquisition part (Page 12, the second paragraph): **“The total time for capturing one 4D data was 122 seconds.”**

(2) How the authors adjust the focus of the probe size during 4D data processing, will the difference in focus at different positions affect the experimental results?

In our experiment, we utilized a 0.5 mrad convergent beam. Therefore, our beam is relatively parallel compared to conventional ADF-STEM imaging, which usually utilize a beam with > 20 mrad convergence angle. The utilization of the small convergence angle results in a relatively large depth of focus that

diminishes the sensitivity of the method to changes in the sample's z height. This is because the depth of focus (D_0) is inversely proportional to the square of the convergence angle.

$$D_0 = \frac{0.61\lambda}{\alpha^2}$$

Where λ is the electron wavelength and α is the convergence angle of the electron beam. In our case, we used the Titan microscope at 300 keV, whose wavelength is around 0.002 nm, and with our utilization of a 0.5 mrad convergence angle, our depth of focus is around 4.88 μm , which is much larger than the existing sample surface fluctuation or TEM holder's drift in z direction in our experiments. Consequently, it is unnecessary to modify the probe's defocus as the diffraction pattern remains minimally impacted during a nanobeam 4D data collection. For comprehensive calculations, please refer to our detailed Method section. We also cite this additional paper to clarify our data acquisition conditions to avoid future confusion².

We revised the paragraph of data acquisition (Page 12, the second paragraph):

“A 0.5-mrad convergence angle was used, leading to a ~ 2.44 nm probe size (defined by Full Width Half Maximum (FWHM) probe diameter) and 4.88 μm depth of focus² (Fig. S27).”

We also added the Figure S27 to clarify our experimental condition:

Figure S27 | EMPAD data acquisition condition. (a) Conventional STEM with a large convergence angle, leading to a smaller probe and a reduced depth of focus. **(b)** Nanobeam STEM with a small convergence angle, resulting in a larger probe and an increased depth of field. **(c)** Schematic illustrating the probe size and depth of field. The probe size, d , is given by $0.61\lambda/\alpha$, where λ represents the wavelength of the electron beam, and α is the convergence angle. The depth of focus, D_0 , is calculated by $d/\tan\alpha \approx 0.61\lambda/\alpha^2$. For our experimental conditions, we utilized a 0.5 mrad convergence angle and a 300 keV electron beam. As a

result, the probe size is 2.44 nm (defined by Full Width Half Maximum (FWHM) probe diameter), and the depth of field is 4.88 μm .

(3) How to judge and correct the scanning distortion?

The scanning distortion caused by sample drift is not a significant concern in our case for the following reasons: Despite an acquisition time of ~ 2 minutes, our data was collected with a large field of view (FOV). The data in **Figure 1** and **Figure 2** were taken at 37k magnification with a 2.6 μm FOV, and the data in **Figure 3** and **Figure 4** were taken at 75k magnification with a 1.3 μm FOV. Our holder's maximum drift is a few nanometers per minute, which is not noticeable at the magnifications we employed. Therefore, unlike the atomic-resolution imaging, the scanning distortion caused by sample drift doesn't significantly impact our nanobeam 4D-STEM data.

We added the following sentence to our Method (Page 12, second paragraph): “In addition, the data in **Figure 1** and **Figure 2** were taken at 37k magnification with a 2.6 μm FOV, and the data in **Figure 3** and **Figure 4** were taken at 75k magnification with a 1.3 μm FOV.”

(4) Will the surface fluctuation of TEM grid affect the experimental results?

The surface fluctuation of the TEM grid does not pose an issue due to the large depth of focus previously described. For the flakes we investigated, we detected no blurry in our virtual ADF images (**Fig. 1c**) from being out of focus, indicating that the sample consistently stayed within the in-focus range.

(5) What is the detection thickness limit of 4D data processing?

So far, our data from SnSe flakes (10-20 nm thick) have not presented any challenges for 4D-STEM analysis. Generally, the detection thickness limit of 4D-STEM is determined by the ability to collect a clear diffraction pattern from very thick samples. To illustrate, we would like to use our data for another project as an example. **Fig. R1** shows 4D data collected during a liquid cell experiment, where the sample thickness exceeds 300 nm. The diffraction pattern is primarily influenced by the solution, and we were not able to analyze the diffraction from the Ag prism. In contrast, for our SnSe sample, which are relatively thin (within 20 nm) with well resolved diffraction patterns, we are not concerned about this issue.

Figure R1 | Thickness limitation in 4D-STEM. (a) ADF image of a 20 nm Ag prism in >300 nm thick liquid within the SiN-window liquid cell. (b) Averaged diffraction pattern from the sample (orange box in (a)). The signal is predominantly from the liquid, and diffraction from the Ag cannot be resolved.

3. The author claimed a large uniaxial strain (4%) in region 2, are there any other experimental evidences, such as HAADF images?

As suggested by the reviewer, we conducted atomic resolution HAADF-STEM imaging in the T-junction area and utilized geometric phase analysis (GPA) to obtain local strain information. The result confirms a ~4% uniaxial strain at the T-junction. These new results have been incorporated into **Fig. S13** and **Fig. S14** of our revised manuscript.

Figure S13 | Strain measurements from atomic resolution ADF-STEM. (a) Low-magnification ADF image of T-junctions. (b) Atomic resolution ADF image of a T-junction, labeled in the white box in (a). (c) Ferroelastic strain map, calculated by geometric phase analysis (GPA) from the atomic resolution image in (b). The strain difference is 4.21% comparing the deformed T-junction and intact neighboring region. This

aligns with the strain measured from 4D-STEM. The detailed calculation of strain using GPA method is shown in **Fig. S14**.

Figure S14 | Comparison between 4D-STEM and GPA strain mapping. (a) Strain maps (ϵ_{xx} , ϵ_{yy} , shear, and rotation) are generated from the 4D-STEM data using a conventional method¹. $[110]$ and $[-110]$ vectors are chosen as the basis for strain matrix calculation. (b) Strain maps of a T-junction from GPA. The uniaxial and shear strain maps are consistent with the 4D-STEM results. (c) The ferroelastic strain map calculated using our method based on 4D-STEM. (d) The ferroelastic strain determined from the shear map in the GPA results, which represents the diagonal deformation (for further details, refer to **Fig. S28**).

4. Even though locating in the “T” shaped junction area (Figure 2c), the second vertical stripe from right to left has a narrow width at the intersection, which is very similar to the needle tip.

We apologize for any confusion caused. As the reviewer pointed out, it is true that the critical width ratio between needle tips and T-junctions is not strictly followed. Instead, there exists a transition zone where both configurations can coexist due to local fluctuations. To clarify this phenomenon, we have revised our manuscript to indicate that there exists a transition region between two configurations.

We revised our main manuscript (Page 5 first paragraph): “We observed that T-junctions are favored when the width ratio is small. Conversely, as the width ratio increases, needle tips become more prevalent. A

transition zone exists between these two configurations from 2.5 to 3 (Fig. 2f). This observation is consistent across multiple samples (Fig. S10), which indicates its ubiquity in vdW SnSe thin flakes.”

Figure 2 | Deformations at superdomain junctions. (a) Lattice schematic of a twin wall. The original lattice without twinning is in dashed blue lines. (b) The difference of diffraction patterns from neighboring stripe domains, showing ~5° rotation. (c,d) Ferroelastic strain (c) and rotation (d) maps at a superdomain boundary where both needle tip morphology and “T” junctions appear. Scale bar: 200 nm. (e) Line profiles of the continuous lattice rotation from the black and blue arrows in (d). (f) Statistics of deformed length depending on the in-plane width ratio (b/a) of the stripe domains. Inset: schematics of the measurements on widths (a and b) and deformed lengths (d).

5. The deformed length, d , is obtained by measuring the distance between the horizontal boundary and the tip of the vertical stripe according to the schematics of Fig. 2f. How is the deformed length actually measured, according to the diffraction spots or the image contrast in Figure 2c-d?

We appreciate the question raised by the reviewer. In response, we have included a new supplementary figure that demonstrates the precise method used to measure the deformed length. To clarify, the deformed

length is determined based on the image contrast of the rotation maps, which are derived from measurements of diffraction spots.

Figure S9 | Deformation length measurement. (a-b) Rotation maps of T-junctions with their intensity line profiles (in the horizontal direction) at various locations, as indicated by the red arrows to the left of the rotation maps. Four lines are displayed here as examples. The horizontal dotted lines indicate the point where the deformation becomes unobservable. The measured deformed length is marked by vertical black arrows, spanning from the intersection to the horizontal line where the deformation fades. (c) Rotation map and intensity line profile of superdomain boundaries with needle tips. As the needle tip structure rarely deforms the horizontal stripes, we measure the gap between needle tips (dotted line) and the intersection. We define this gap as the deformed length (indicated by vertical black arrows). Since the direction of deformation in needle tips is opposite to that in T-junctions, the deformed length in the needle tip structure is considered negative.

We revised the main text to incorporate the changes (Page 5 last paragraph): “We measured the deformed length, d , of each vertical stripe at the boundary (as shown in the inset schematics of Fig. 2f) by the image contrast in Fig. 2d (details in Fig. S9). The deformed length is positive when a_1^- deforms a_1^+ and generates “T” junctions (Fig. S9 a,b), and negative when a gap is observed in between, which usually form needle tips (Fig. S9 c).”

6. The author claimed that the critical width ratio is around 2.7. What’s the physical mechanism of this value? Can the authors discuss further with the relation to the in-plane strain? Did authors conduct phase field in this state?

We apologize for any confusion caused. As mentioned in our response to Question 4, we wish to clarify that the critical width ratio represents a transition zone between the two configurations, meaning the

transition is not abrupt. Following the reviewer’s suggestion, we conducted additional phase field simulations, particularly around the transition zone (3:1 and 4:1). These revealed that the underlying physical mechanism involves the elastic energy variation within the horizontal stripes, affecting the strain propagation across the superdomain boundaries. We have incorporated these results into the newly added **Fig. S11**. In addition, we collected more experimental data, which confirmed the consistency of this phenomenon across different flakes. This additional data is illustrated in **Fig. S10**.

We made the following changes to our manuscript:

Figure S11 | Phase field simulation of superdomain boundaries with different width ratios. (a) Rotation map showing domains with varying width ratios from 1:1 to 6:1. **(b)** Simulated maps of the elastic free energy density. Notably, the elastic energy in the horizontal domains (highlighted in the red boxes) increases as the domain width ratio rises. When the elastic energy of the horizontal domain is low, it is more susceptible to deformation by vertical stripes. In contrast, higher elastic energy resists the deformation, leading to the formation of the needle tip structure. The transition zone is observed around 3:1 width ratio. **(c)** Simulated maps of the electrostatic energy density, indicating that the electrostatic energy remains unaffected by the domain width ratio.

We added more data of “T junction” and “Needle tips” structures from the additional flakes in **Fig. S10**.

Figure S10 | T-junctions and needle tips from additional flakes. (a) Low magnification ADF image of the Flake I. (b) Zoomed-in ADF image of region 1 in Flake I. (c-d) Ferroelastic strain and rotation map of region 1. Typical examples of T-junctions are shown in the dotted box. (e) Zoomed-in ADF image of region 2 in Flake I. (f-g) Ferroelastic strain and rotation map of region 2. Additional examples of T-junctions are highlighted in the dotted box. (h) Low-magnification ADF image of another flake (Flake II). (i) Zoomed-in ADF image of a selected region in Flake II. (j-k) Ferroelastic strain and rotation map of the region shown in (i). In addition to examples of T-junctions (in the dotted box), we also observed numerous needle-tip structures, attributable to a relatively wide vertical stripe spanning the field of view.

We further discuss the relation between the transition zone and in-plane strain (page 6 first paragraph): “We observed that T-junctions are favored when the width ratio is small. Conversely, as the width ratio increases, needle tips become more prevalent. A transition zone exists between these two configurations from 2.5 to 3 (Fig. 2f). This observation is consistent across multiple samples (Fig. S10), which indicates its ubiquity in vdW SnSe thin flakes.”

7. What is the distribution law of horizontal and vertical stripes width? Can the width of stripes be actively controlled? For example, changing the annealing time, temperature?

(1) What is the distribution law of horizontal and vertical stripes width?

The stripe width distribution is mostly from 20 nm to 80 nm. We included the histogram of the stripe width distribution from multiple flakes to a new **Figure S7**.

Figure S7 | Histogram of the domain width on the as grown SnSe flake. (a) The flake in the main manuscript. (b) Flake I in Fig. S1. (c) Flake II in Fig. S2.

We revised our manuscript to clarify this point (Page 4 second paragraph): “...observed intertwined domain morphology and a domain width that mostly ranged from 20 to 80 nm in thin-flake SnSe (**Fig. S7**).”

(2) Can the width of stripes be actively controlled? For example, changing the annealing time, temperature?

Currently, we cannot control the stripe width during PVD synthesis. However, we are investigating effective ways to regulate the stripe width. Our preliminary results indicate that fast cooling leads to the formation of narrower stripe domains (**Fig. R2**). This points to a potential way to control stripe width by

varying annealing conditions. We plan to include these results in a separate study focused on domain width control and phase transition.

Figure R2 | Domain width before and after fast quenching. (a) Low-magnification ADF image of an SnSe flake before a heating and fast cooling process. (b) Zoomed-in ADF image of the red box region in (a). (c) Ferroelastic strain map of the selected region. The domain width is around 45 nm. (d) Low-magnification ADF image of the same flake after fast cooling. (e) Zoomed-in ADF image of the same region in (b) and (c). (f) Ferroelastic strain map of the selected region after fast cooling. The domain width is around 12 nm.

8. The author analyzed two types of the morphologies: needle tips and “T” shaped junctions. In the practical devices, what are the specific application advantages of the two superdomain boundaries?

These findings are important for practical application in two aspects.

(1) These two morphologies will lead to different local deformations at the superdomain boundaries, which will affect their switching energy. Specifically, needle tips often display nonuniform widths and encompass a more pronounced lattice rotation that incrementally rotates the polarization. In contrast, T-

junctions exhibit a more consistent shape, but with a 4% tensile strain in the box 2 area. This suggests an intensified polarization within a nanometer-scale region, accompanied by minimal lattice distortion in adjacent areas. The significant differences in deformation types will influence the switching energy, thus affecting their viability in memory and actuator device applications.

(2) The presence of two morphologies is affected by the domain size (stripe width), indicating a pathway to control morphology through domain width engineering. For example, a uniform domain width predominantly results in 1:1 ratio, fostering the dominance of T-junction superdomain boundaries. In contrast, a more pronounced variation in domain width yields both T-junction and needle tips at superdomain boundaries. These insights provide valuable direction for synthesis teams, guiding them in engineering domain sizes to achieve desired structures for specific applications.

We revised our conclusion to highlight the practical advantages of the two morphologies (Page 9 second paragraph): “The "T" shaped junction and needle-tip morphology we identified in vdW FEs have profound implications, especially in influencing switching energy and their viability in memory and actuator device applications. Our insights into domain width dependence pave the way for strain control through domain size engineering.”

9. The authors simulated diffraction patterns of 10-unit-cell thick SnSe lattices that contain different ratios of AC and AB stacking orders using the kinetic diffraction theory model in Figure S8. Why the intensity of two spots at the lower right corner changes more obviously than the two spots at the upper left corner?

The intensity difference of the $\{110\}$ spots are from the sequence of the AB stacking lattice. For example, the A-B stacking and the B-A stacking will show the opposite intensity of the conjugate $\{110\}$ spots. We added a new Figure S17 to illustrate the effect.

Figure S17 | Multislice simulation of diffraction patterns from pure AC and AB stacking. (a-b) AC stacking with the opposite polarization directions. (c-d) AB stacking with sequences “A-B” and “B-A”. In the “A-B” sequence (c), the $\{110\}$ spots on the left are brighter than those on the right, while the opposite is true in the “B-A” sequence (d). To reduce the errors from this unbalanced intensity, we sum the intensities of the four $\{110\}$ spots.

10. The authors utilized the atomic model and diffraction spots of FE structure (AC) to investigate the in-plane strain in Figure 1-3. But the intensity of $\{110\}$ spots ($\sim 1.4 \times 10^4$) in the majority area is obvious, corresponding to the AFE (AB) stacking ordering. Will this fact affect the in-plane strain results?

This will not affect the in-plane strain results. We revised our manuscript to clarify this point (Page 12 second paragraph): “A camera length of 720 mm was employed to ensure clear separation between the $\{110\}$ spots and the $\{200\}$ spots. This distinction allows our strain mapping (based on the $\{200\}$ spots) and stacking mapping (based on the $\{110\}$ spots) to be entirely independent.”

11. The authors utilized the normalized intensity ($\{110\}$ intensity /total intensity of the diffraction pattern) to investigate the stacking sequence variation. What does the “total intensity” refer to?

The total intensity refers to the sum of the intensity in the entire diffraction pattern, representing the approximate total number of incoming electrons.

We add the following text in the Method to clarify this point (Page 14 paragraph 1): “To quantify the stacking order and provide a comparison consistent to simulations, we normalized the summed intensity of the $\{110\}$ spots by dividing it by the integrated intensity of the entire diffraction pattern. We opted not to

utilize the {200} spot intensity for normalization since it is more sensitive to variations caused by lattice tilts.”

12. The authors mentioned that Figure S10 is the normalized intensity, but it is clear that the range of Y axis is much larger than 1. Also, what does the X axis stand for?

We apologize for the confusion. We forgot to include the labels for the x and y axes. The Y-axis represents the counts of pixels from our 4D data, and the X-axis denotes the normalized intensity. We have addressed this issue in the previous Figure S10 (now Fig. S19).

13. What’s the parameter of the ADF-STEM image in Figure 4d? What’s the simulation condition for the inset simulated images?

We now included the ADF-STEM imaging condition and the simulation in our Method (page 15, second paragraph).

“**ADF-STEM imaging and simulation:** ADF-STEM image is taken in 300kV, and the dwell time is 10 μ s at each scan position. The convergence angle of the electron probe is 25 mrad, with a 115 mm camera length. The simulated ADF-STEM images are generated by the multi-slice method in abTEM³ package with the same conditions.”

Reviewer #2 (Remarks to the Author):

In this manuscript, by utilizing 4D-STEM, the authors probed the microscopic strain and rotation in 2D SnSe to visualize the complicated crystal structures around the domain walls. By this means, the authors could directly quantify the deformation and rotation of the crystals. They found strain as large as 4% which could exist in a 50 nm narrow domain wall region and a width-dependent T junction at the 180° domain walls. Meanwhile, the authors also verified the unique ferroelectric-antiferroelectric domain structures arising from different out-of-plane stacking. Overall, this research is of current interest and the manuscript is well organized. However, a few critical questions should be addressed before any recommendation can be given.

We thank the reviewer for finding our work interesting.

1. Through the manuscript, the authors only characterized one sample and even one region on the sample?

The reviewer understands that the measurements might be challenging; however, is this sufficient to draw a general conclusion?

We appreciate the reviewer's question. To address this concern, we conducted additional measurements on two more SnSe thin flakes and incorporated the data into our new supplementary figures (**Fig. S1** and **Fig. S2**). While the previous data primarily focused on one sample, we have now observed that the domain structure, local deformation, and stacking orders are consistent across other samples as well.

Figure S1 | 4D-STEM measurements from an additional flake (Flake I). (a) ADF-STEM images of an entire single-crystalline SnSe thin flake, with two zoomed-in regions for detail. (b) Corresponding ferroelastic strain maps. (c) Rotation maps. (d) (110) intensity maps indicating the stacking order.

Figure S2 | 4D-STEM measurements from a different flake (Flake II). (a) ADF-STEM images of another single-crystalline SnSe thin flake, with two zoomed-in regions for detail. (b) Corresponding ferroelastic strain maps. (c) Rotation maps. (d) (110) intensity maps indicating the stacking order.

2. May be related to the comment 1, as the authors only characterized one sample, will the method and/or process of the preparation affect the domain structures? For example, since the AB and AC stacks in SnSe nanoflake represent different phases, would the mixed FE-AFE orders in SnSe nanoflake form due to the specific method and/or process?

The reviewer inquired about the potential influence of sample preparation procedures on the domain structure. As discussed in R1 Q7 (2), we've demonstrated that rapid cooling can alter domain size (**Fig. R2**). Additionally, we noted that annealing can lead to changes in the stacking order when domains reconfigure, as shown in **Figure R3**. We aim to conduct a systematic exploration of domain tuning and phase transitions in SnSe thin flakes. We believe this investigation merits its own dedicated publication; therefore, we did not incorporate the results into our current manuscript.

As for other synthesis parameters, such as growth temperature, pressure, and synthesis duration, we have not yet identified a correlation with the domain structure. Even though we have tested various conditions –

temperatures ranging from 370°C to 470°C, pressures between 340 mTorr and 1830 mTorr, and growth durations from 5 minutes to 4 hours – our synthesis paper⁴ highlights that these factors mainly affect variations in flake size and thickness. In contrast, the domain structure in SnSe thin flakes remained consistent across all examined conditions.

However, it is worth noting that the thickness of SnSe flakes significantly influences domain formation. Thicker SnSe flakes (>50 nm) typically lack a domain structure and predominantly exhibit the AFE phase (or AB stacking). The thickness-dependency of group-IV monochalcogenides synthesized via PVD have been discussed in these articles^{4,5}.

Figure R3 | Normalized [110] intensity maps showing the stacking order before (left) and after (right) a heating and fast cooling process.

3. Basically, the as-prepared 3D ferroelectric materials always contain various domain structures. What is the main feature of the domain structures probed in this manuscript. Is it possible to tune the domain structures?

(1) What is the main feature of the domain structures probed in this manuscript.

We thank the reviewer for this insightful question. In our study, the domain structure of SnSe thin flakes aligns with the subgrain twin domains observed in 3D perovskite ceramics^{6,7}. However, they also exhibit

distinct features that differ from 3D domain structures. Two primary unique characteristics emerge in our vdW SnSe FE thin flakes:

- Given the flexible lattice structure inherent to 2D SnSe, our SnSe flakes can accommodate significantly large lattice strains—up to 4% based on our findings—without causing bond breakage or forming dislocations or grain boundaries. As a result, our larger flakes (>10 μm) maintain their single crystalline nature.
- Our observations revealed a coexistence of FE and AFE domains, leading to the formation of unique FE-AFE domain walls in vdW FEs. This phenomenon is attributed to the weakly-bonded vdW layered structures, allowing for the formation of stable head-to-head or tail-to-tail domain walls in certain layers.

These distinctive properties differentiate vdW FEs from their bulk counterparts, offering fresh perspectives on the understanding and potential applications of vdW FEs.

We noticed that the confusion of the reviewer may come from our title. Therefore, we changed our title to “**Domain-dependent Strain and Stacking in Two-dimensional van der Waals Ferroelectrics**” to highlight the key findings from our work.

We also revised our abstract: “However, their electric polarization is strongly coupled with the lattice strain and stacking orders, ~~resulting in complex domain structures with lattice distortions and stacking effects that~~ **which significantly impact their electronic properties and domain wall switching.**”

We revised our introduction: “The limited experimental results have motivated new approaches to revealing the **strain and stacking across various domains** in ultrathin MX compounds, particularly the heterogeneity **manifested at distinct domain wall types.**”

(2) Is it possible to tune the domain structures?

Regarding the tuning of domain structures, we have addressed a similar question earlier (R1, Q7, (2)). Our preliminary results (**Fig. R2**) indicate that the domain size can be tuned by cooling conditions. We are currently engaged in a systematic study to understand the phase transition and domain control. We intend to incorporate these findings in our subsequent manuscript.

4. Generally, the in-plane deformation includes rotation, normal strain and shear strain. Could the definition of strain and rotation in this work effectively differentiate these contributions? For example, will the shear strain also contribute to the rotation as defined?

We appreciate the reviewer's question. We added the following paragraph in our method to address this question (Page 13 second paragraph): “We devised this mapping strategy in response to the limitations of conventional measurements from the strain matrix, which often fail to clearly delineate polarizations and their angles in complex, intertwined domains. In the conventional method¹, the four strain maps (e_{xx} , e_{yy} , shear, and rotation) are derived based on a selected basis (Fig. S28). While it's theoretically possible to compute the ferroelastic strain and armchair rotation from the shear and rotation maps produced by the conventional method (see Fig. S29), the presence of deformations at superdomain boundaries, such as the continuous lattice rotation, complicates the decoupling of shear and rotation, often yielding imprecise results. In contrast, our method directly assesses the ferroelastic strain and rotation from diffraction patterns, producing more accurate maps.”

We included a new SI figure to show the strain maps from conventional polar decomposition approach:

Figure S28 | Strain maps from the polar decomposition approach. (a) Strain maps using [200] and [020] as the basis vectors. The e_{xx} and e_{yy} maps show the uniaxial strain along [200] and [020]. The e_{xy} map shows minimal contrast, indicating the [200] and [020] vectors remain mostly orthogonal. The rotation map illustrates the lattice rotation of the two vectors. **(b)** Strain maps utilizing [110] and [-110] as the basis vectors. The e_{xx} and e_{yy} maps exhibit minimal contrast, indicating minor strain along [110] and [-110]. The e_{xy} map shows the diagonal deformation, which is along [200] and [020]. Therefore, the e_{xy} map shows the

similar contrast with the ferroelastic map, the detailed explain in **Figure S29**. The rotation map remains consistent with the one in (a).

We included another SI figure to explain how we decouple the shear strain from the lattice rotation using out approach:

Figure S29 | Relationship between shear strain and ferroelastic strain. (a) Schematic of the SnSe lattice during phase transition. SnSe lattices can elongate in either the [200] direction or the [020] direction. (b) Schematic illustrating the relationship between shear strain and ferroelastic strain. Due to lattice elongation along the diagonal ([200]) direction, the unstrained (or squared) shape lattice transforms into a rhombus-shaped lattice. This transformation induces a small shear angle, α , as labeled in (b). Given that the uniaxial strain along x ([110]) and y ([-110]) directions is closed to zero (**Fig. S28b**) and α is small:

$$\alpha \approx \sin\alpha = \frac{\Delta x}{y} = \frac{\Delta y}{x} = \frac{1}{2} \left(\frac{\Delta x}{y} + \frac{\Delta y}{x} \right) = e_{xy},$$

where e_{xy} is the shear strain. Meanwhile, the [200] lattice constant can be expressed as:

$$d_{200} = x \times \cos\left(\frac{\pi}{4} - \alpha\right) \times 2 = x \times \left(\frac{\sqrt{2}}{2} \times \cos\alpha + \frac{\sqrt{2}}{2} \sin\alpha \right) \times 2 = \sqrt{2} x (\cos\alpha + \sin\alpha)$$

Since α is small, $\cos\alpha \approx 1$, $\sin\alpha = \alpha$. Therefore, the diagonal lattice constants are:

$$d_{200} = \sqrt{2} x (1 + \alpha) = \sqrt{2} x (1 + e_{xy})$$

$$d_{020} = \sqrt{2} x (1 - \alpha) = \sqrt{2} x (1 - e_{xy})$$

And the ferroelastic strain is:

$$\frac{d_{200}}{d_{020}} - 1 = \frac{1 + e_{xy}}{1 - e_{xy}} - 1 = \frac{2e_{xy}}{1 - e_{xy}} \approx 2e_{xy}$$

In terms of lattice rotation, we utilized the “armchair rotation”, which refers to the rotation angle of the elongated diagonal lattice direction. As a result, the two cases presented in (b) exhibit a $\sim 90^\circ$ rotation. In contrast, the conventional polar decomposition strain mapping method (**Fig. S28**) cannot extract this information.

5. Although a few studies have reported the ferroelectric properties of 2D SnSe, were it better if the authors also characterize, at least briefly, ferroelectric properties of their flakes. After all, ferroelectric properties depend on thickness, stacking and so on... In particular, the ferroelectric and antiferroelectric regions are claimed to coexist.

We followed the reviewer’s suggestion and performed additional characterization of ferroelectric properties of our flake. Here we provided additional PFM measurements, second-harmonic generation (SHG) data, and FE device measurements. The PFM measurements reveal the stripe twin domain structure and provide insights into their polarization. We provide the results in our new **Figure S3**. We also provided new SHG data (**Fig. S25**) show angle-dependent polarization at different domains, confirming the coexistence of FE and AFE domains and their influence of stacking order on nonlinear optical responses of vdW SnSe. Specifically, the FE phase is noncentrosymmetric while the AFE phase is centrosymmetric. According to group theory, the FE region will exhibit nontrivial second-harmonic generation (SHG), but the AFE region will have vanishing/minimal SHG response. This is exactly what was observed in **Fig. S25** with alternating strong and low SHG response regions, demonstrating coexisting FE and AFE regions. In addition, we incorporated the results of FE device measurements of vdW SnSe thin flakes from a recent publication⁷ using the same batch of sample we investigated (**Fig R4**). The results demonstrate the electrical switching of our SnSe thin flakes and their hysteresis loop.

Figure S3 | Atomic force microscopy (AFM) and piezoresponse force microscopy (PFM) images of a SnSe thin flake. (a) AFM image showing the surface topography of a SnSe thin flake on its original substrate (mica surface). **(b-c)** The amplitude and phase images from PFM measurements, revealing the stripe twin domain structure within the SnSe flake, offering insights into its polarization. **(d-e)** Zoomed-in amplitude and phase images from PFM measurements which show the stripe twin domain structures.

Figure S25 | SHG images of as-grown SnSe thin flakes. (a-b) SHG intensity maps, taken at different angles, from an as-grown SnSe on a mica substrate, showing areas with reduced intensity (red boxes). These areas indicate predominant AB stacking (AFE phase). **(c-d)** SHG intensity maps from a different as-grown SnSe flake on mica. The regions (red box) with diminished SHG intensity indicate areas of AB stacking (AFE phase). In contrast, areas with high SHG intensity regions correspond to AC stacking (FE phase). The observations confirm that as-grown SnSe inherently exhibits a coexistence of both AB and AC stacking.

Figure R4 | Electrically switching SnSe. (a) Optical image of the SnSe device with Pb/Cr electrodes and 400, 800, 1200, 2000 nm channels. Scale bar is 500 nm. (b) Q-E hysteresis loop of ~10 nm SnSe. These figures are cited from the recent paper⁸ by our coauthor Nannan Mao and Jing Kong.

1. Han, Y. *et al.* Strain Mapping of Two-Dimensional Heterostructures with Subpicometer Precision. *Nano Lett* **18**, 3746–3751 (2018).
2. Hovden, R., Xin, H. L. & Muller, D. A. Extended Depth of Field for High-Resolution Scanning Transmission Electron Microscopy. *Microsc. Microanal.* **17**, 75–80 (2010).
3. Madsen, J. & Susi, T. The abTEM code: transmission electron microscopy from first principles. *Open Res Europe* **1**, 24 (2021).
4. Chiu, M. *et al.* Growth of Large-Sized 2D Ultrathin SnSe Crystals with In-Plane Ferroelectricity. *Adv Electron Mater* 2201031 (2023) doi:10.1002/aelm.202201031.
5. Higashitarumizu, N. *et al.* Purely in-plane ferroelectricity in monolayer SnS at room temperature. *Nat Commun* **11**, 2428 (2020).
6. Xiao, X. *et al.* Benign ferroelastic twin boundaries in halide perovskites for charge carrier transport and recombination. *Nat. Commun.* **11**, 2215 (2020).
7. Hermes, I. M. *et al.* Ferroelastic Fingerprints in Methylammonium Lead Iodide Perovskite. *J. Phys. Chem. C* **120**, 5724–5731 (2016).
8. Luo, Y. *et al.* Electrically switchable anisotropic polariton propagation in a ferroelectric van der Waals semiconductor. *Nat Nanotechnol* 1–7 (2023) doi:10.1038/s41565-022-01312-z.

REVIEWERS' COMMENTS

Reviewer #1 (Remarks to the Author):

The authors have addressed my comments in a satisfactory manner with changes and additions in the manuscript.

Reviewer #2 (Remarks to the Author):

In this revised version, the authors have addressed most questions properly. The revised manuscript can now be accepted.